# Exploring kinetically induced bound states in triangular lattices with ultracold atoms: spectroscopic approach

Ivan Morera[1*], Christof Weitenberg[2,3], Klaus Sengstock[2,3,4] and Eugene Demler[1]

**1** Institute for Theoretical Physics, ETH Zurich, 8093 Zurich, Switzerland
**2** Institut für Quantenphysik, Universität Hamburg, 22761 Hamburg, Germany
**3** The Hamburg Centre for Ultrafast Imaging, 22761 Hamburg, Germany
**4** Zentrum für Optische Quantentechnologien, Universität Hamburg, 22761 Hamburg, Germany

* imorera@ethz.ch

December 4, 2023

## Abstract

Quantum simulations with ultracold fermions in triangular optical lattices have recently emerged as a new platform for studying magnetism in frustrated systems. Experimental realizations of the Fermi Hubbard model revealed striking contrast between magnetism in bipartite and triangular lattices. In bipartite lattices magnetism is strongest at half filling, and doped charge carriers tend to suppress magnetic correlations. In triangular-type lattices for large $U/t$, magnetism gets enhanced by doping away from $n = 1$ because kinetic energy of dopants can be lowered through developing magnetic correlations. This corresponds to formation of magnetic polarons, with hole and doublon doping resulting in antiferro- and ferromagnetic polarons respectively. Snapshots of many-body states obtained with quantum gas microscopes [1–3] demonstrated existence of magnetic polarons around dopants at temperatures that considerably exceed superexchange energy scale. In this paper we discuss theoretically that additional insight into properties of magnetic polarons in triangular lattices can be achieved using spectroscopic experiments with ultracold atoms. We consider starting from a spin polarized state with small hole doping and applying a two-photon Raman photoexcitation, which transfers atoms into a different spin state. We show that such magnon injection spectra exhibit a separate peak corresponding to formation of a bound state between a hole and a magnon. This polaron peak is separated from the simple magnon spectrum by energy proportional to single particle tunneling and can be easily resolved with currently available experimental techniques. For some momentum transfer there is an additional peak corresponding to photoexciting a bound state between two holes and a magnon. We point out that in two component Bose mixtures in triangular lattices one can also create dynamical magnetic polarons, with one hole and one magnon forming a repulsive bound state.

# 1   Introduction

Spectroscopic probes have become a powerful tool for unraveling quantum correlations in many-body systems [4]. Techniques such as inelastic neutron scattering [5], xray scattering [6], and angle resolved photoemission spectroscopy (ARPES) [7] have been used in traditional solid-state experiments to measure single electron Green's functions, spin and charge response functions with both frequency and momentum resolution. These studies have provided crucial insights into several families of quantum materials, including heavy fermion systems, topological and low dimensional materials, copper and iron based superconductors [8, 9]. Experimental progress of atomic physics in the last few decades has allowed to develop similar spectroscopic techniques for exploring many-body systems realized with ultracold atoms [10]. Radiofrequency spectroscopy and its finite momentum extension have been used to probe strongly interacting Fermi gases across the BCS-BEC crossover regime [11–16]. Ramsey interferometry has been succesfully applied to probe spin dynamics and transport in Fermi and Bose gases [17–23] and to extract the contact parameters [24]. Moreover, two-photon Bragg spectroscopy has allowed to measure excitation spectra of weakly and strongly interacting Bose and Fermi gases [25–32]. Lattice modulation has been used to measure excitation spectra of bosons in optical lattices of different geometry [33–37]. One of the key tasks of spectroscopic experiments is to understand emergent quasiparticles that appear as a result of many-body correlations. In particular, Fermi and Bose polarons have been investigated in cold

atom systems via radiofrequency spectroscopy [38–47], ARPES [48] and two-photon Raman spectroscopy [49]. However, polarons that arise from the interplay of charge and spin degrees of freedom, i.e. magnetic polarons, remain less explored experimentally using spectroscopic techniques. Magnetic polarons provide an important starting point for understanding unusual properties of high-$T_c$ superconductors [50–54]. On the theoretical side, most efforts have been focused on understanding magnetic polarons in square lattices, with similar results expected to hold for all bipartite lattices [50, 52–68]. However several theoretical studies considered non-bipartite triangular lattices and pointed out intriguing effects of frustration on magnetic polarons [69–73]. Specifically, antiferromagnetic polarons appear when a Mott insulator in a triangular lattice is doped with a low density of holes. Antiferromagentic spin correlations allow to alleviate some of the kinetic energy loss suffered by holes on a triangular lattice. The characteristic binding energy of such composite objects is determined by the tunneling strength and not the superexchange interaction. This makes them promising candidates for experimental studies with cold atoms simulators, which are still limited in how low they can go in temperature [69, 70]. In particular, recent cold atom experiments have simulated the Hubbard model in the triangular lattice [1–3, 74–76] and explored the emergence of kinetic magnetism with fermionic atoms loaded in triangular geometries [1]. Moreover, they also have reported the observation of antiferromagnetic and ferromagnetic polarons [2, 3]. Furthermore, several theoretical studies have also pointed out the importance of understanding magnetic polarons in triangular lattices for deciphering the rich physics of transition metal dichalcogenide (TMDc) moiré materials [72, 73, 77, 78]. Recent experiments provided strong evidence for both kinetic ferromagentism [79] and antiferromagentism [80, 81] in these materials. The central goal of this work is to analyze how spectroscopic techniques of ultracold atoms can be employed to detect kinetically induced magnetic polarons in weakly doped Mott insulators in frustrated triangular geometries. We consider systems close to full spin polarization and present calculations for two specific lattice geometries: the triangular ladder and the two-dimensional triangular lattice.

In this paper we focus on systems in which atoms are fully polarized before a two-photon Raman excitation and only briefly comment on the more general case. During the pulse, a small number of atoms are transferred into a different spin state, with most of the photoexcited atoms becoming independent magnon excitations. There is also a finite probability that a spin flip process occurs in the vicinity of a hole and results in formation of a magentic polaron. The unambiguous experimental signature of magnetic polaron formation is the appearance of a new peak in the photoexcitation spectrum. At low hole doping, the amplitude of this peak should be proportional to the hole density. The energy difference between the free magnon and magnetic polaron peaks is given by the binding energy of a magnetic polaron, which is of the order of the tunneling strength. Our calculations show that higher order multi-body composites can also be detected spectroscopically, including the possibility of observing bipolarons formed by two holes. However, experimental constraints on the observation of bipolarons are more demanding. The weight of the bipolaron peak is proportional to the square of the hole density, which is parametrically smaller, and one needs to employ two-photon Raman processes with momentum transfer in a smaller range. Considering the remarkable frequency resolution of optical spectroscopy, we expect that magnon, polaron, and bipolaron peaks should all be resolvable in currently available experimental systems. We present calculations for fermionic and bosonic systems, and show that fermions (bosons) feature attractively (repulsively) bound antiferromagentic polarons. This means that for bosons the polaron peak appears at a frequency higher than that of a free magnon. Additionally, we show that bosons are subject to a parity selection rule, making the antiferromagnetic polaron a dark state with respect to the two-photon Raman process, when one starts with a hole doped Mott insulator. To circumvent this problem, we present a two-stage protocol, in which holes are first accelerated to a finite

momentum before using a two-photon Raman pulse to create spin excitations with a similar momentum transfer.

This paper is organized as follows. We begin by reviewing the microscopic models studied in our work in Sec. 2. We review spectroscopic measurements relevant for our analysis in Sec. 3. We present the main results in Sec. 4 and review the numerical methods employed in Sec. 5. Our microscopic calculations for the fermionic and bosonic models are presented in Sec. 6 and Sec. 7, respectively. We also discuss different experimental considerations in Sec. 8 Finally, we present conclusions in Sec. 9

## 2   Microscopic models

We consider a system of spin-1/2 particles moving in a triangular lattice with a hopping amplitude $t$. We consider both fermionic and bosonic systems. The respective Hamiltonians are given by,

$$\hat{H}_F = -t \sum_{\langle ij \rangle \sigma} \left( \hat{c}_{i\sigma}^\dagger \hat{c}_{j\sigma} + \text{h.c.} \right) + U_{\uparrow\downarrow} \sum_i \hat{n}_{i\uparrow} \hat{n}_{i\downarrow}, \tag{1}$$

$$\hat{H}_B = -t \sum_{\langle ij \rangle \sigma} \left( \hat{b}_{i\sigma}^\dagger \hat{b}_{j\sigma} + \text{h.c.} \right) + \frac{U}{2} \sum_{i\sigma} \hat{n}_{i\sigma} \left( \hat{n}_{i\sigma} - 1 \right) + U_{\uparrow\downarrow} \sum_i \hat{n}_{i\uparrow} \hat{n}_{i\downarrow}, \tag{2}$$

where $\hat{c}_{i\sigma}^\dagger$ and $\hat{b}_{i\sigma}^\dagger$ creates a fermion and a boson at site $i$ with spin $\sigma = \uparrow, \downarrow$, respectively, and $\hat{n}_{i\sigma}$ is the number operator. We assume that both types of atoms tunnel with equal hopping strength $t$ and we introduce on-site strengths for both cases $U_{\sigma\sigma'}$. Moreover, we introduce the lattice constant $a = 1$.

In our calculations we exploit the underlying U(1)⊗U(1) symmetry of Hamiltonians (1) and (2) associated with particle number conservation in each spin sector. Therefore, we introduce the total number of particles in each spin state $N_\uparrow$ and $N_\downarrow$, respectively. Moreover, we define the number of holes in the system as $N_h = N_s - (N_\uparrow + N_\downarrow)$, being $N_s$ the total number of sites in the lattice.

## 3   Setup

We start by considering an insulating state of fermionic atoms with $\downarrow$ spin which has some defects (holes) on top of it, see Fig. 1a). We focus on the regime where the number of holes is small compared to the system size $N_h/N_s \ll 1$. When a hole propagates in a kinetically frustrated lattice, including a triangular lattice, it experiences kinetic frustration due to the destructive interference between different propagation trajectories [69, 70, 82–84]. This obstruction to kinetic energy of a hole can be alleviated if the hole binds a spin flip excitation. Resulting decrease of the hole kinetic energy underlies formation of a magnetic polaron. When a $\downarrow$ atom is photoexcited into the $\uparrow$ state, there is a finite probability that this magnon creation process takes place in the vicinity of a hole and results in formation of a hole-magnon bound state. Thus, the photoexcitation spectrum exhibits the following features: one peak corresponding to the free magnon creation and a second peak at a smaller frequency given by the hole-magnon bound state. When $U_{\sigma\sigma'}$ is finite, we also have corrections to binding energy being just $t$. So the condition is that all $U_{\sigma\sigma'}$ should be large. When repulsion is strong, i.e. all $U_{\sigma\sigma'}/t \gg 1$, energy difference between the two peaks is proportional to $t$. To determine

energies of individual polarons that are not modified by polaron-polaron interactions, spectroscopy should be performed at low hole densities $n_h \ll 1/(z+1)$, where $z$ is the coordination number of the lattice, i.e. the number of nearest neighbors of a single site.

The setup that we consider can be extended to systems that are not fully polarized before the pulse. In this case it is possible to probe multi-body bound states comprised of more than one magnon. Consider a Mott insulator of $\downarrow$ atoms with a small density of both, holes and $\uparrow$ atoms. When photoexciting a $\downarrow$ atom into the $\uparrow$ state, there is a finite probability of doing so in the vicinity of a hole-magnon bound state. In this case, a trimer state formed by two magnons and a hole can be created. The probability of this event is proportional to the density of hole-magnon bound states, given by $\min(N_h, N_\uparrow)$. In this way, multi-body bound states can be probed, resulting in multiple peaks in the photoexcitation spectrum. Identifying higher order composite objects spectroscopically faces two challenges. Firstly, the difference in energy between the 2M1H (two magnons/one hole) composite vs 1M1H (one magnon/one hole) and one free magnon is smaller than the binding energy of a single polaron. Hence the peak corresponding to formation of the 2M1H state will be at a smaller frequency. Secondly, the amplitude of this peak is expected to be small due to the reduced probability of photoexciting an atom in the vicinity of a large composite object.

The photoexcitation spectrum $I_\beta(\vec{k}, \omega)$ at momentum $\vec{k}$, frequency $\omega$ and temperature $\beta = 1/(k_B T)$ can be related to the spin structure factors,

$$I_\beta(\vec{k}, \omega) = \frac{\Omega^2}{\mathcal{Z}} \lim_{\eta \to 0} \text{Im} \sum_{m,n} e^{-\beta \omega_m} \left[ \frac{\left| \langle n | S_{\vec{k}}^+ | m \rangle \right|^2}{\omega - \omega_m + \omega_n + i\eta} - \frac{\left| \langle n | S_{\vec{k}}^- | m \rangle \right|^2}{\omega + \omega_m - \omega_n - i\eta} \right], \qquad (3)$$

where we introduce the partition function $\mathcal{Z}$ and the strength of the light-matter coupling $\Omega$. Note that the second term vanishes for initial configurations with no $\uparrow$-atoms. To access finite momentum properties one can consider a two-photon Raman process with coupling strength $\Omega$, see Fig. 1c).

# 4 Overview of results

Before presenting our microscopic calculations we summarize the main results of our work.

## 4.1 Spectroscopy of fermions in a triangular ladder

Fermionic holes propagating in the triangular ladder suffer from kinetic frustration. Simultaneously, magnons also experience kinetic frustration due to the antiferromagnetic superexchange coupling, resulting in the dispersion relations of free magnons and holes having two degenerate minima at finite momentum $\pm k_0 a = \pm 2 \arctan \sqrt{5/3}$, resembling a two valley structure, see Fig. 1b). To alleviate the kinetic frustration, holes and magnons form a hole-magnon bound state which has a dispersion relation with a minimum at zero momentum, see Sec. 6. This corresponds to a low energy hole-magnon bound state composed of a hole and a magnon in the same valley.

Net momentum of the photoexcited hole-magnon bound state is a sum of the initial momentum of the hole and momentum imparted by light. At low temperatures we expect holes to have momenta close to $\pm k_0$. Therefore to get a zero momentum bound state, we need Raman pulse to give a momentum to the system equal to $\pm k_0$. When the system is probed at these momenta, we observe a two-peak structure in the photoexcitation spectrum corresponding to the free magnon and the hole-magnon bound state, see Fig. 1f). Moreover, the two peaks are separated by a frequency given by the hole-magnon binding energy which is of the order of $t$. When approaching the edge of the first Brilluoin zone, a third peak appears

in the spectrum below the hole-magnon dispersion relation, denoting formation of a trimer state composed by two holes and the photoexcited magnon, see Fig. 1g). This composite object appears in a narrow range of momenta and is separated from the hole-magnon dispersion relation by a frequency $\sim 0.2t$ at $k = \pm\pi$. Remarkably, the photoexcitation spectrum reveals a change in the groundstate properties of the three-body problem as a function of the net momentum. In Sec.6.1, we confirm the appearance of a trimer state at $k = \pm\pi$ by employing non-Gaussian states with momentum resolution. Larger bound states composed of multiple magnons can also be probed by starting with configurations containing some initial magnons and holes. The photoexcitation spectrum exhibits different peaks corresponding to the formation of multi-magnon bound states, see Sec. 6.2. Finally, we discuss the temperature effects on the photoexcitation spectrum in Sec. 6.3. At temperatures $k_B T \leq t$, a sharp peak develops below the free magnon dispersion relation. This peak corresponds to the hole-magnon bound state and it is separated from the free magnon peak by the respective binding energy. For larger temperatures, the peaks are smeared and therefore, they are more difficult to resolve.

### 4.2 Spectroscopy of bosons in a full two-dimensional triangular lattice

In Sec. 7, we present results for the bosonic system in a full 2D triangular lattice. An interesting feature of bosonic systems is that they exhibit an anti-bound state (repulsively bound state) composed by a hole and a magnon. Observing this anti-bound state spectroscopically requires utilizing a more elaborate measurement protocol. First, a potential gradient must be employed to accelerate a hole to momentum $P_y$. Then, a two-photon Raman process should be applied to create a magnon excitation with similar momentum. Hence this protocol allows to create a hole-magnon bound state with null net momentum. The optimal momentum for producing the anti-bound state corresponds to $P_y$ close to the K point of the triangular lattice. As we discuss below, this two-stage protocol is required because a simple photoexcitation with zero momentum transfer has a vanishing matrix element to couple to the anti-bound state.

## 5 Numerical methods

Before presenting our main calculations we provide details on the different methods employed in our work.

### 5.1 Tensor Networks

#### 5.1.1 Real time evolution

To obtain the zero temperature photoexcitation spectrum presented in Fig. 1 we Fourier transform the time evolution of the correlation function [85],

$$S^+(k, \omega) = \int dt \sum_j e^{ikja - iwt} \langle\psi_0|e^{i\hat{H}t} S_j^- e^{-i\hat{H}t} S_0^+|\psi_0\rangle, \tag{4}$$

which gives us access to the dynamical spin structure factor $S^+(k, \omega)$. The ground state $|\psi_0\rangle$ is computed using the density matrix renormalization group (DMRG) algorithm on a matrix product state (MPS) with maximum bond dimension $\chi = 1000$. Then, we flip a spin in the middle of the lattice and time evolve the system using the time evolving decimation block (TEBD) algorithm with a maximum bond dimension $\chi = 2000$ and a finite time step $\delta t = 0.05$.

### 5.1.2 Lanzcos algorithm

The dynamical spin structure factor at zero temperature can be expressed in the Lehmann representation as,

$$S^+(\vec{k}, \omega) = \sum_n |\langle n|S_{\vec{k}}^+|0\rangle|^2 \delta(w - E_0 - E_n), \tag{5}$$

where we introduce the eigenstates of the Hamiltonian $|n\rangle$ and their eigenenergies $E_n$, being $n = 0$ the ground state. To numerically obtain Eq. (5) we need to obtain the set of eigenvectors with a non-vanishing overlap with the vector $S_{\vec{k}}^+|0\rangle$. To efficiently create the set of eigenvectors we iteratively generate a Krylov space where the starting vector is given by $S_{\vec{k}}^+|0\rangle/|S_{\vec{k}}^+|0\rangle|^2$. Details on the implementation of the Lanzcos algorithm can be found in [86]. The Lanzcos algorithm provides the discrete set of spectral weights $\Omega_{n,\vec{k}} = |\langle n|S_{\vec{k}}^+|0\rangle|^2$ and poles $E_n$. To obtain a continuum version of the dynamical spin structure factor we employ a Lorentzian broadening,

$$S^+(\vec{k}, \omega) = \frac{1}{\pi} \sum_n \Omega_{n,\vec{k}} \frac{\eta/2}{(w - E_n)^2 + (\eta/2)^2}. \tag{6}$$

The photoexcitation spectrum presented in Fig. 3 is obtained performing the aforementioned Lanzcos algorithm using MPS with maximum bond dimension $\chi = 1000$. We keep 120 Lanzcos vectors to compute the dynamical spin structure factor and we discard the contributions to the spectrum coming from states with a spectral weight smaller than $\Omega_{n,\vec{k}} < 10^{-4}$. We also make sure that the autocorrelations in our Krylov space remain of the order of $10^{-6}$ during the construction of the Lanzcos vectors.

## 5.2 Non-Gaussian states

The results presented in Fig. 2 are obtained by restricting ourselves to the three-body problem of a single magnon and two holes. We first perform a Lee-Low-Pines (LLP) transformation to the reference frame of the magnon and then propose a Gaussian wavefunction for the two holes which we optimize by performing an imaginary time evolution [87]. The Hamiltonian (1) after LLP transformation reads,

$$\hat{H}_{\vec{K}} \equiv \hat{U}_{\text{LLP}}^\dagger \hat{H} \hat{U}_{\text{LLP}} = t \sum_{\vec{k}, \vec{\delta}} e^{-i\vec{k}\cdot\vec{\delta}} \hat{h}_{\vec{k}}^\dagger \hat{h}_{\vec{k}} - t \sum_{\vec{\delta}} e^{-i(\vec{K} - \hat{Q}_h)\cdot\vec{\delta}} + U_{\uparrow\downarrow} - \frac{U_{\uparrow\downarrow}}{N_s} \sum_{\vec{k}, \vec{q}} \hat{h}_{\vec{k}}^\dagger \hat{h}_{\vec{q}}, \tag{7}$$

where we introduce hole operators $\hat{h}_{\vec{k}} = \hat{c}_{\vec{k},\downarrow}^\dagger$, the hole momentum operator $\hat{Q}_h = \sum_{\vec{k}} \vec{k} \hat{h}_{\vec{k}}^\dagger \hat{h}_{\vec{k}}$ and the lattice vectors $\vec{\delta} = \{\pm a, \pm 2a\} \cdot \hat{e}_x$. The Gaussian hole wavefunction is described by the correlation matrix $\Gamma_{\vec{k},\vec{q}} = \langle \hat{h}_{\vec{k}}^\dagger \hat{h}_{\vec{q}} \rangle_{\text{GS}}$ which we optimize to represent the ground state of the system by performing an imaginary time evolution,

$$d_\tau \Gamma = 2\Gamma h \Gamma - \{h, \Gamma\}. \tag{8}$$

We introduce the matrix $h_{\vec{k},\vec{q}} = \frac{\delta E[\Gamma]}{\delta \Gamma_{\vec{k},\vec{q}}}$ and the energy of the system computed using our Gaussian state $E[\Gamma] = \langle H_{\vec{K}} \rangle_{\text{GS}}$,

$$E[\Gamma] = \sum_{\vec{k},\vec{q}} \epsilon_{\vec{k}} \Gamma_{\vec{k},\vec{q}} \delta_{\vec{k},\vec{q}} + U_{\uparrow\downarrow} - \frac{U_{\uparrow\downarrow}}{N_s} \sum_{\vec{k},\vec{q}} \Gamma_{\vec{k},\vec{q}} - t \sum_{\vec{\delta}} e^{-i\vec{K}\cdot\vec{\delta}} \det\left[1 + \left(e^{-i\alpha} - 1\right)\Gamma\right], \tag{9}$$

where we define the matrix $\alpha = \text{diag}(\vec{k} \cdot \vec{\delta})$ and $\epsilon_{\vec{k}} = \text{diag}\left(\sum_{\vec{\delta}} e^{-i\vec{k}\cdot\vec{\delta}}\right)$. To simulate the imaginary time evolution given by Eq. (8) we employ a Runge-Kutta algorithm of fourth order with a discrete time step $\delta\tau = 0.01$ and obtain the ground state with an energy convergence $\delta E/\delta\tau = 10^{-4}$.

The imaginary time evolution Eq. (8) conserves the purity of a Gaussian state $d_\tau \left(\Gamma^2 - \Gamma\right) = 0$ and therefore, the total number of holes $N_h = \text{Tr}(\Gamma)$ is also conserved in the evolution $d_\tau \text{Tr}(\Gamma) = 0$. However, the conservation laws may not be satisfied in the imaginary time evolution due to numerical errors leading to instabilities in the evolution. To fix this problem we propose an additional minimization problem which restricts the time evolution in the subspace of pure Gaussian states. We introduce the Lagrangian,

$$\mathcal{L}\left[\bar{\Gamma}\right] = \frac{1}{2}||\bar{\Gamma} - \Gamma||^2 - \lambda\left(\bar{\Gamma}^2 - \bar{\Gamma}\right), \tag{10}$$

that describes a new correlation matrix $\bar{\Gamma}(\tau)$ at each imaginary time step which has a minimum distance with $\Gamma(\tau)$ and is forced to be idempotent $\tilde{\Gamma}^2 = \tilde{\Gamma}$. The equation of motion $\delta\mathcal{L} = 0$ leads to the solution,

$$\tilde{\Gamma} \sim \Gamma - \lambda\left(2\Gamma - 1\right), \tag{11}$$

where we have assumed that $\lambda$ is small. The Euler-Lagrange multiplier is found in a self-consistent way by imposing the idempotent property $\tilde{\Gamma}^2 = \tilde{\Gamma}$ which we solve using the Newton-Raphson algorithm,

$$\lambda_{i+1} - \lambda_i = \frac{-1}{(2\Gamma - 1)^2}\left(\Gamma^2 - \Gamma\right), \tag{12}$$

where the subindex $i$ denotes the step in the algorithm. By choosing an initial value of lambda $\lambda_0 = 0$ we can make the algorithm efficient and we can rapidly determine the new pure correlation matrix $\bar{\Gamma}$ given a quasi-pure correlation matrix $\sum_{\vec{k},\vec{q}}\left(\bar{\Gamma}_{\vec{k},\vec{q}}^2 - \bar{\Gamma}_{\vec{k},\vec{q}}\right) = \epsilon$. In practice, we keep the correlation matrix pure under imaginary time evolution with an error $\epsilon = 10^{-4}$. In this way, we make sure that the number of holes is conserved and avoid instabilities during the evolution.

## 6  Fermions in a triangular ladder

Holes in the triangular ladder feature a dispersion relation with two degenerated minima at finite momenta $\pm k_0 a = \pm 2\arctan\sqrt{5/3}$ forming a two valley structure. The holes are equally distributed around these two minima for sufficiently low temperatures. Thus, a photoexcited a magnon can be bound with a hole in any of the two valleys. Therefore, we expect to observe two different branches corresponding to bound states $|\Phi_Q^{\pm}\rangle$ with total momentum $Q$. Note that the total momentum is related to the hole and magnon momentum as $Q = k_m - k_h$. For this geometry we obtain the following dispersion relations for the bound states at $U_{\uparrow\downarrow}/t \gg 1$ close to the minimum of each valley,

$$\omega_\pm(k_m) \approx E_B + \frac{t}{2}(1 - \cos(k_m \mp k_0)) - \frac{J}{3}\left(\cos(k_m \mp k_0)\right) + 2\cos\left(2(k_m \mp k_0)\right) - 1), \tag{13}$$

where $E_B \approx -0.735t - 3.125J$ is the hole-magnon binding energy, $J = 4t^2/U$ is the superexchange interaction and we assume that the bound hole is located in one of the two valleys $k_h = \pm k_0$. On the other hand, the free magnon has a similar dispersion relation to that of a free hole,

$$\omega_m(k_m) = J\left(\cos(k_m) + \cos(2k_m) - 2\right). \tag{14}$$

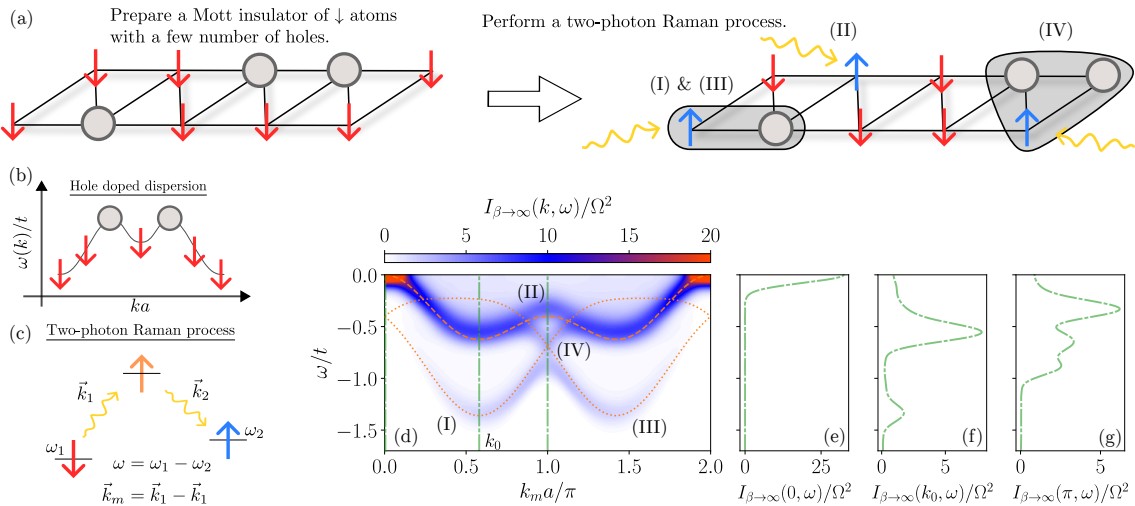

Figure 1: A Mott insulator of ↓ atoms doped with a small number of holes is prepared in a triangular ladder, see panel (a). Then, a two-photon Raman process is performed (see panel (c)) and a ↓ atom is photoexcited into the ↑ state. When the photoexcited atom is far away from the holes it would have a large overlap with the free-propagating magnon state, see (II). On the other hand, when the photoexcitation is produced close to a hole we will have a large overlap with the bound state, see (I) and (III). Since the groundstate wavefunction of holes is at momentum $\pm k_0$ (see panel (b)) we expect two different types of bound states. The photoexcited magnon can bind either with a hole at $+k_0$ or at $-k_0$. Moreover, a trimer state can be formed if the photoexcited magnon is close to two holes, see (IV). This state only appears close to the edge of the Brillouin zone. On panel (c) we show the two-photon Raman process employed to flip the spin of a particle i.e. create a magnon. On panel (d) we show the photoexcitation spectrum as a function of the momentum and frequency of the magnon. The calculation is performed in a triangular ladder with $N_s = 70$ sites and $N_h = 4$ holes with an interaction $U_{\uparrow\downarrow}/t = 20$. Note that we plot the first Brilluoin zone from 0 to $2\pi$ just for convenience. Dashed line corresponds to the free ↑-fermion dispersion relation, see Eq. (14). Dotted lines correspond to the hole-magnon bound state dispersion relations, see Eq. (13) corresponding to the binding of the photoexcited ↑-fermion with the hole at momentum $\pm k_0$. A new peak (IV) appears below the pair branch (I or III) at the edge of the Brillouin zone denoting the formation of a trimer state of two holes together with the photoexcited magnon. On panels (e), (f) and (g) we show cuts of the photoexcitation spectrum at fixed momentum $k_m = 0, k_0, \pi$, respectively. At $k_m = 0$ there is a single peak corresponding to the free-magnon propagation. At $k_m = k_0$ a second peak appears denoting the formation of a hole-magnon bound state. At $k_m = \pi$ the third peak corresponding to the trimer can be resolved.

Therefore, the hole-magnon bound state is separated by a frequency $\omega \sim t$ from the free magnon dispersion relation at $k_m = \pm k_0$.

To obtain the photoexcitation spectrum we compute the spin structure factor at zero temperature using tensor network methods, see Sec. 5.1. In Fig. 1c) we present the photoexcitation spectrum as a function of the probed frequency $\omega$ and momentum $k_m$. Three main branches can be identified in the spectrum. The upper one (II) corresponds to the dispersion relation of a free magnon. This branch appears when the photoexcitation is produced far away from a hole. However, if the photoexcitation occurs in the vicinity of a hole then we have a finite

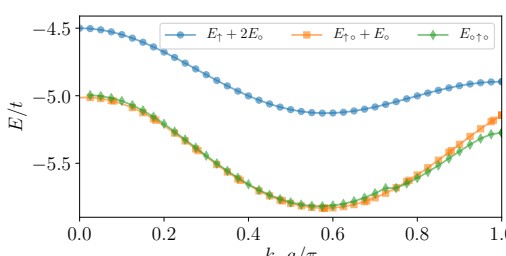 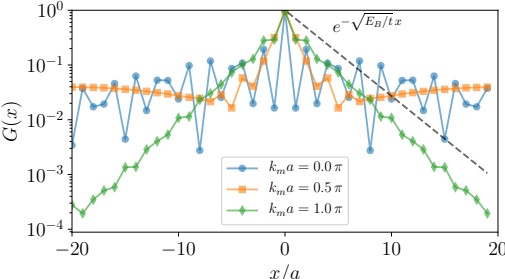

Figure 2: Left panel: Dispersion relation as a function of the photoexcited magnon momentum $k_m$ for a free spin flip, a pair and a three body state for an interaction strength $U_{\uparrow\downarrow}/t = 20$ in a lattice of $N_s = 40$ sites. Right panel: Hole function $G(x)$ as a function of the relative distance for different photoexcited momenta $k_m$ computed with the same parameters than in the top panel. Dashed line shows an exponential decay with a decay length given by the square root of the binding energy $E_B \sim 0.13t$ at $k_m a = \pm\pi$.

overlap with the hole-magnon bound state. The lower branches (I) and (III) correspond to this process and the corresponding frequencies are given by the dispersion relations of the bound states, see Eq. (13). The two branches can be distinguished by the probed momentum $k_m$. Branch (I) (or (III)) corresponds to binding between the photoexcited magnon and a hole with momentum $k_h = k_0$ (or $k_h = -k_0$). However, the total quasi-momentum $Q = k_m - k_h$ of the hole-magnon bound state is not a good quantum number for a system with more than a single hole. Therefore, we expect a mixing between the two bound states as we discuss below.

## 6.1 Spectroscopically revealed hole pairs

The dispersion relations of the two hole-magnon bound states intersect each other at the edges of the Brillouin zones $k_m a = \pm\pi \pm 2\pi n$, see Eq.(13). At these points the magnon has a momentum equidistant to the hole in one valley or the other. Thus, it can equally bind to any of the two holes. The photoexcitation spectrum shows an avoided crossing instead of a direct one at this momentum, see Fig. 1c), signaling the appearance of a matrix element connecting the two different bound states. The way to understand this effective interaction is to introduce a second free hole into the picture and construct the states $|\Phi^+_{k_m - k_h}\rangle \otimes |-k_h\rangle$ and $|\Phi^-_{k_m + k_h}\rangle \otimes |k_h\rangle$. A scattering process between the bound state and the free hole is possible: the bound state with total momentum $Q = k_m - k_0$ can scatter with the hole in the other valley $-k_0$ and then interexchange the momenta of the two holes. Thus, the two states are connected and an effective pair-hole interaction appears. This interaction is enhanced close to the edges of the Brillouin zones where the bare states become degenerated. The photoexcitation spectrum (see Fig. 1c)) for a system doped with few holes shows two peaks below the free magnon branch (II) at $k_m a = \pm\pi$: one is located at a frequency given by the free pair and hole energies and the other one appears at a smaller frequency. The appearance of a new peak signals the formation of a new state, a trimer formed by two holes and the photoexcited fermion. It may seem surprising that we observe a trimer bound state (two holes and one magnon) at all in this system, since there are lower energy dimer (one hole and one magnon) states. As we wil show with non-Gaussian states, the dimer and hole state have higher energy compared with a trimer at the same total momentum (close to the edge of the Brilluoin zone).

The photoexcitation spectrum has shown an unexpected new peak appearing at the edge of the Brillouin signaling the appearance of a new composite object formed by two holes and one

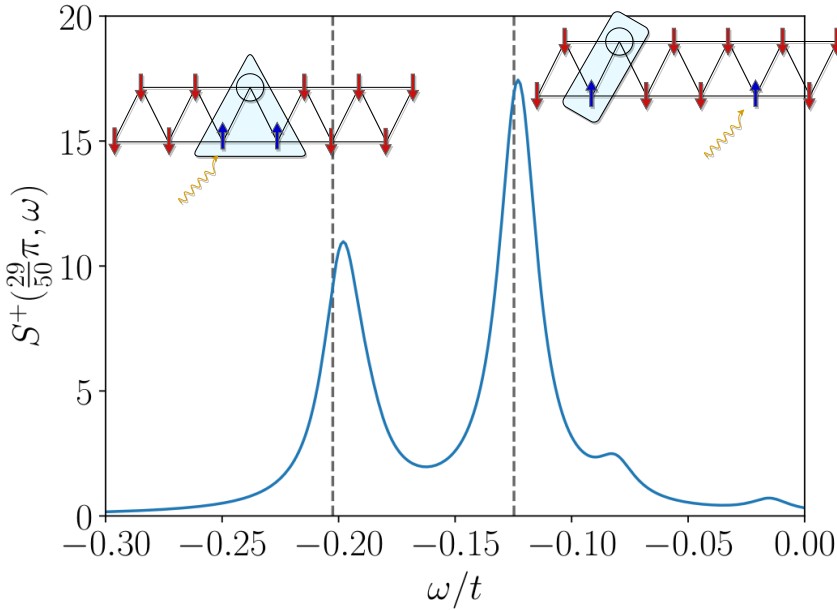

Figure 3: Spin structure factor $S^+$ at a magnon momentum $k_m = 29\pi a/N_s$ as a function of the probed frequency $\omega$. The calculation is performed in a triangular ladder with $N_s = 50$, $N_h = 1$ and $N_m = 1$ with an interaction $U_\uparrow/t = 100$. A Lorentzian broadening has been applied with a characteristic scale $\eta = 1/N_s$. When the photoexcited fermion is far away from the holes and magnons we recover the free propagation. This corresponds to the peak centered at frequency $\omega/t \approx -0.125$. On the other hand, if the photoexcited fermion is close to a hole and another magnon we will have an overlap with the trimer. This corresponds to the second peak centered at frequency $\omega/t \approx -0.203$. The trimer is formed by a hole and a magnon already present in the system and the photoexcited magnon.

magnon. In order to benchmark this prediction we are going to solve the three-body problem by employing Gaussian states together with a Lee-Low-Pines transformation allowing us to solve the three-body problem with a fixed total momentum of the system. Experimentally, this net momentum comes from the momentum transferred in the two-photon Raman process. In Fig. 2 we show the dispersion relations as a function of $k_m$ for a single spin flip, a hole-magnon pair and a three-particle state. By comparing the energies at the same net total momentum we observe that the trimer becomes lower in energy than a dimer plus a free hole close to the edge of the Brillouin zone $k_m a \gtrsim 3/4$. In this region the two holes are bound by an effective interaction mediated by the magnon. In order to confirm the bound nature of the state we compute the correlation function of the two holes $G(x) = \langle \hat{c}_{i\uparrow} \hat{c}_{i\uparrow}^\dagger \rangle$, where $x = |i-j|a$. As shown in Fig. 2 the correlation function saturates at large distances for small probed momentum $k_m$. However, the correlation function decays exponentially fast near the edge of the Brilluoin zone $k_m \sim \pm \pi$. This is a clear indicator of the bound state nature of the two holes for this momentum. Moreover, we observe that the decay length of the exponential decay is given by the binding energy of the trimer state which becomes $E_B \sim 0.13t$ at $k_m a = \pm\pi$.

## 6.2 Trimers

As shown above the photoexcitation spectrum can also denote the appearance of multiparticle bound states formed by a photoexcited magnon and multiple holes. We can employ the same method to observe a different type of bound states in which there are several magnons and one

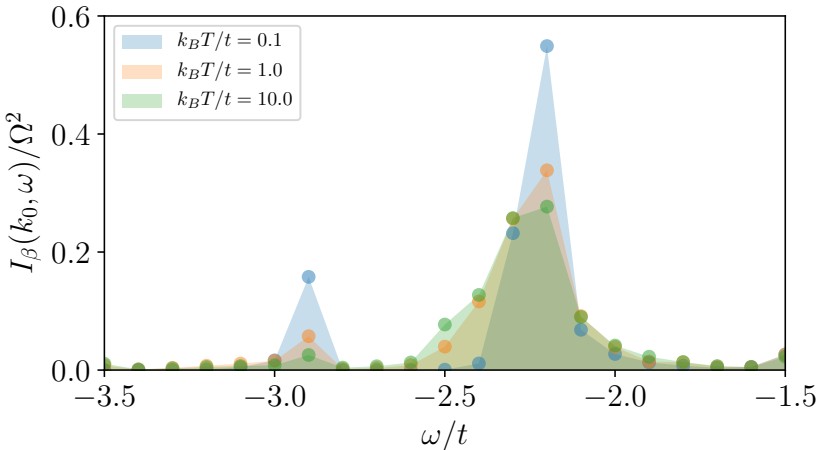

Figure 4: Photoexcitation spectrum for a fermionic system in a triangular ladder at momentum $k_m = k_0$ as a function of the probed frequency for different temperatures. The system contains a single hole in $N_s = 101$ sites with an interaction strength $U_{\uparrow\downarrow}/t = 20$. The peak at smaller frequency denotes formation of the hole-magnon bound state and the other one denotes formation of the free magnon state. The pair peak is sharp for temperatures lower than the tunneling strength and it is smeared out as the temperature increases.

hole. The simplest one will correspond to a trimer formed by a single hole and two magnons. In order to observe this multiparticle bound state we have to reach an equilibrium state with some magnons and holes on top of the $\downarrow$ insulator. Then, when a photoexcitation creates an additional magnon, we can access bound states that consist of two or more magnons. In Fig. 3 we present the spin structure factor $S^+$ for a system with a single hole and a single $\uparrow$-fermion. We observe that two peaks appear in the spectrum for large systems: one peak corresponding to the free magnon state and another peak at smaller frequencies corresponding to the trimer state. The two peaks are separated by a characteristic frequency corresponding to the binding energy of the trimer. Notice that this binding energy corresponds to the channel of a trimer decomposing into a free hole-magnon bound state (dimer) plus a free magnon. Therefore, the trimer binding energy is smaller than the dimer one but still it can be of the order $\sim -0.1t$.

## 6.3 Finite temperature effects on the photoexcitation spectrum

Temperature plays an important role in the photoexcitation spectrum since it provides a major source of smearing. In order to quantify this effect we compute the finite temperature photoexcitation spectrum in the two-particle sector, see Eq. (16). The peak denoting the pair bound state is smeared by increasing temperature, see Fig. 4. Since the binding energies of our problem are of the order of the tunneling strength $t$, the system must be cooled down to temperatures below $t$ to see sharp peaks in the spectrum.

Most of our discussion has focused on the fermionic system in a triangular ladder. However our results also apply to the full two-dimensional triangular geometry. The main difference is that hole pairs do not appear in the 2D geometry with the Hamiltonian considered. However multiparticle states formed by multiple magnons are expected.

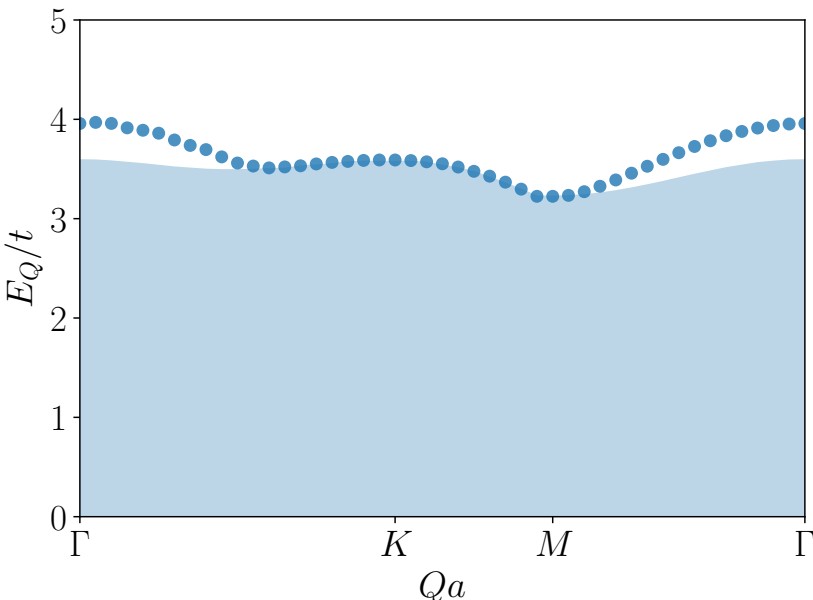

Figure 5: Dispersion relation as a function of total quasi-momentum $Q$ of a two-particle state formed by a hole and a magnon in the effective bosonic $t-J$ model for a triangular lattice $40 \times 40$ sites with an interaction stregth $U_{\uparrow\downarrow}/t = 40$. Dots denote the anti-bound state and the blue filled region represents the scattering continuum.

## 7   Bosons in triangular lattice

Let us now study the bosonic system in a triangular lattice with Hamiltonian (2). In this situation we have repulsive bound states of holes and magnons i.e. states that appear above the scattering continuum. In order to solve the two-body problem we first derive the effective bosonic $t-J$ model,

$$\hat{H} = -t \sum_{\langle ij \rangle \sigma} \left( \hat{b}^{\dagger}_{i\sigma} \hat{b}_{j\sigma} + \text{h.c.} \right) + \frac{J_{\perp}}{2} \sum_{\langle ij \rangle} \left( S^{+}_{i} S^{-}_{j} + \text{h.c.} \right) + J_{z} \sum_{\langle ij \rangle} S^{z}_{i} S^{z}_{j}$$

$$- \frac{2t^2}{U} \sum_{\langle ijk \rangle \sigma} \left( \hat{b}^{\dagger}_{i\sigma} \hat{n}_{j\sigma} \hat{b}_{k\sigma} + \text{h.c.} \right) - \frac{t^2}{U_{\uparrow\downarrow}} \sum_{\langle ijk \rangle \sigma} \left( \hat{b}^{\dagger}_{i\sigma} \hat{n}_{j\bar{\sigma}} \hat{b}_{k\sigma} + \hat{b}^{\dagger}_{i\sigma} S^{\bar{\sigma}}_{j} \hat{b}_{k\sigma} + \text{h.c.} \right), \qquad (15)$$

where $J_{\perp} = 4t^2/U_{\uparrow\downarrow}$, $J_z = 4t^2/U_{\uparrow\downarrow} - 8t^2/U$ and we introduce the notation $S^{\uparrow}_{i} = S^{+}_{i}$ and $S^{\bar{\uparrow}}_{i} = S^{-}_{i}$. The first term in the bosonic $t-J$ model allows the holes to hop with a strength $t$. While the second term and third term denote the effective superexchange magnetic interactions with strengths $J_{\perp}$ and $J_z$, respectively. Moreover we include the three-site terms. The first one allows to holes to hop up to next-nearest neighbors and the other ones are effective interactions between the hole and the $\uparrow-$boson. These last terms generally appear when holes are present in the system. By simplicity we focus on the SU(2) symmetric point i.e. $U = U_{\uparrow\downarrow}$, but our results could be easily generalized.

We solve the two-particle problem of the effective bosonic $t-J$ model, see Fig. 5. An anti-bound state appears close to the $\Gamma$ point and its binding energy is of order of the tunneling strength $t$. Thus we could expect that the photoexcitation spectrum will signal a peak at the $\Gamma$ point separated by the scattering continuum by a frequency of order $t/2$.

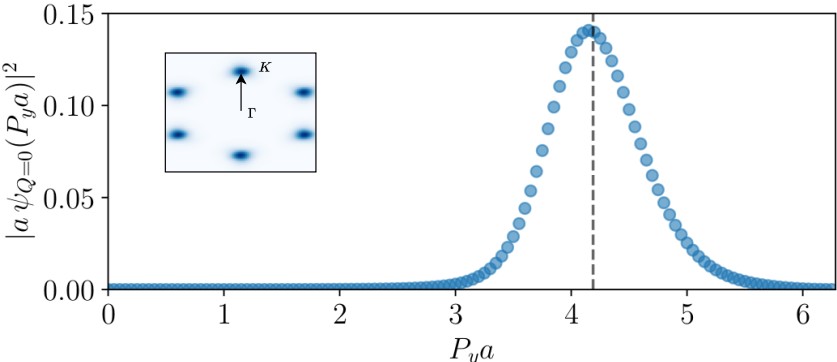

Figure 6: Main panel: Bosonic magnon-hole bound state wavefunction at total quasi-momentum $Q = \Gamma$ as a function of the hole quasi-momentum $P_y$ for an interaction $U = U_{\uparrow\downarrow} = 20$. The dashed line denotes the $K$ point. Inset panel: Magnon-hole bound state wavefunction at total quasi-momentum $Q = 0$ in the first Brilluoin zone.

## 7.1 Parity selection rules: Bright and dark pairs

By solving the two-particle problem of one bosonic hole and one magnon we observe that the energy of the anti-bound pair is maximal at the $\Gamma$ point, see Fig. 5. This can be understood as follows. Without magnons, bosonic holes have a maximal energy at the $K$ and $K'$ points at zero temperature. To create an anti-bound state at zero momentum, we then need to add a magnon with the momentum equal to that of the hole. Moreover, by solving the two particle problem we observe that the bound state wavefunction is odd with respect to the exchange of the hole and the magnon position at zero total momentum. In the triangular lattice this imposes that the bound state should be f-wave symmetric in order to respect the underlying $C_6$ symmetry. Similarly, the bound state is p-wave symmetric in the triangular ladder. Therefore, in both geometries the bound pair is odd under a parity transformation imposing a selection rule for the photoexcitation spectrum. The selection rule can be obtained by examining the expression of the photoexcitation spectrum Eq. (3) in the two particle sector,

$$I_\beta(\vec{k}_m, \omega) = \frac{1}{\mathcal{Z}} \sum_n \sum_{\vec{k}_h \in 1BZ} |\tilde{\psi}^{(n)}_{\vec{k}_m - \vec{k}_h}(\frac{\vec{k}_h + \vec{k}_m}{2})|^2 e^{-\beta w_h(\vec{k}_h)} \delta\left(w - (w^{(n)}_{\vec{k}_m - \vec{k}_h} - w_h(\vec{k}_h))\right), \quad (16)$$

where we introduce the hole dispersion relation $\epsilon(\vec{k}_h)$ and the Fourier transform of the relative wavefunction for the state with total momentum $\vec{Q} = \vec{k}_m - \vec{k}_h$. Since the wavefunction is odd under the exchange of the two particles then $|\tilde{\psi}^{(n=0)}_0(0)|^2 = 0$. Thus the matrix elements connecting to the bound or anti-bound state are zero when we apply pulses with zero momentum in a system with holes at zero momentum. For the fermionic system this is not a problem since at zero temperature holes are not located at zero momentum but at a finite one: $k_h = \pm k_0$ in the triangular ladder and $k_h = K, K'$ in the triangular lattice. Thus, the bound state is bright in this range. However, for bosonic systems the ground state with some holes is located at zero momentum due to the effective change of sign of $t$ respect to fermions. Thus, hole-magnon anti-bound states are dark under a photoexcitation probe. In order to circumvent this problem we present an extended protocol to avoid the parity selection rule.

## 7.2 Pump and probe protocol

To make the anti-bound state bright under photoexcitation we need to change the hole background. Specifically, we have to give it a finite momentum in analogy with the fermionic

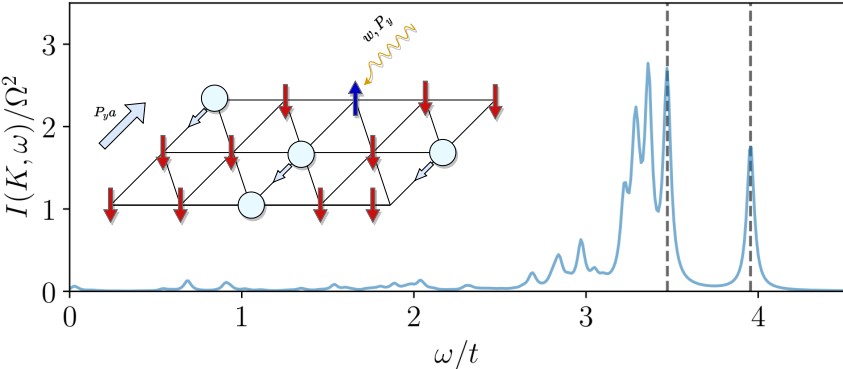

Figure 7: Schematic representation of the pump and probe protocol for probing anti-bound states with spectroscopy. We consider a system with a low concentration of holes inside the insulating state of $\downarrow$ bosons. In equilibrium they primarily occupy states with momenta close to zero. Potential gradient applied to the system accelerates holes to finite momentum $P_y$. Once the holes are accelerated we photoexcite a $\downarrow$-atom into an $\uparrow$-atom using a two-photon Raman process that transfers momentum $P_y$ and energy $\omega$. Main panel: Photoexcitation spectrum $I(K, \omega)$ for a bosonic system with interaction $U = U_{\uparrow\downarrow} = 20$ in a triangular lattice with $12 \times 12$ sites as a function of the probed frequency $\omega$. The spectrum has been computed following the pump and probe protocol proposed. The dashed line at smaller frequency denotes the maximum of the scattering continuum. The second dashed line above the scattering continuum denotes the energy of the two-particle anti-bound state.

case. In Fig. 6 we show the anti-bound state wavefunction at null total quasi-momentum as a function of the hole momentum along the $y$-axis $P_y$. As can be seen, the anti-bound state wavefunction is peaked around the $K$ point signaling that we should give a momentum $K$ to the hole in order to have a finite matrix element and make the anti-bound state bright. At the same time, we want to keep the total quasi-momentum of the anti-bound state to be zero $Q = \Gamma$ to have a large binding energy, see Fig. 5. Thus we need to photoexcite the magnon at a momentum close to the one given to the hole background. This idea can be summarized in the following pump and probe protocol: First accelerate the Bose insulator with some defects and then perform a two-photon process to photoexcite a boson with similar momentum than the one given to the accelerated holes. In Fig. 7 we show the resulting photoexcitation spectrum following the pump and probe protocol for a bosonic system in a triangular lattice. The photoexcitation spectrum shows a peak above the scattering continuum and its frequency coincides with the binding energy of the two particle anti-bound state.

Furthermore, our protocol could also be employed to resolve the dispersion relation of the anti-bound state. A mismatch in the momentum of the holes $\vec{k}_h$ and the photoexcited boson $\vec{k}_m$ will allow us to study the energy dependence of the two-particle state with the total quasi-momentum $\vec{Q} = \vec{k}_m - \vec{k}_h$.

## 8 Experimental considerations

The (anti-)binding energy on the scale of the tunneling $t$ instead of the superexchange coupling $J$ makes these predictions experimentally accessible. The chosen parameter of $U_{\uparrow\downarrow}/t = 20$ yields an enhancement of the relevant energy scale of $t/J = 5$, but they still require large $U_{\sigma\sigma'}$ and $t$. This can be achieved, e.g., in a relatively shallow lattice and enhanced interac-

tions via a Feshbach resonance. The duration of the Raman pulse should be chosen at $1/t$ such that Fourier broadening still allows resolving the spectral features, but no significant dynamics occurs during the pulse. The robustness of the signal to finite temperatures discussed in section 6.3 is quite promising for experimental realizations. Current cold-atom experiments reach temperatures around $k_B T = 0.4t$ for fermionic spin mixtures in triangular lattices [2], which will allow for well-resolved spectra.

For probing the hole-magnon bound states, no precise preparation of the hole density is required. While we consider here a fixed doping as experimentally realized in box traps, a spectroscopic measurement of the bound state should also be possible in harmonically trapped systems. In this case, one could focus the Raman beams onto the system in order to probe only the center of the cloud.

## 9 Conclusions and Outlook

The photoexcitation spectrum of strongly polarized two-component fermionic and bosonic mixtures in lattices with frustrated geometry can reveal the presence of bound and anti-bound states, respectively. We focus on filling factors close to one, allowing us to describe these systems from the perspective of a dilute gas of holes in an insulating state. In this scenario, the photoexcited atom binds to a single hole in order to release the kinetic frustration of the hole. Since the binding energy is of order $t$, the photoexcitation spectrum exhibits two peaks separated by a frequency of order $t$. The peak at a small frequency corresponds to the free magnon propagation, while the other one represents the bound state. Two types of hole-magnon bound states appear in the spectrum and they can be distinguished by the transfer momentum in the photoexcitation process. They correspond to the binding of the magnon with a hole in one of the two valleys. Moreover, the two bound states have an effective coupling mediated by an extra hole in the opposite valley. This effective interaction creates a trimer state composed by two holes and the photoexcited atom close to the edge of the Brillouin zone.

The setup that we consider can be extended to systems that are not fully polarized before the pulse. In this case it is possible to probe multi-body bound states comprised of more than one magnon. Consider a Mott insulator with of ↓ atoms with a small density of both, holes and ↑ atoms. When photoexciting a ↓ atom into the ↑ state, there is a finite probability of doing so in the vicinity of a hole-magnon bound state. In this case, a trimer state formed by two magnons and a hole can be created. The probability of this event is proportional to the density of hole-magnon bound states, given by $\min(N_h, N_\uparrow)$. In this way, multi-body bound states can be probed, resulting in multiple peaks in the photoexcitation spectrum. Identifying higher order composite objects spectroscopically faces two challenges. Firstly, the difference in energy between the 2M1H (two magnons/one hole) composite vs 1M1H (one magnon/one hole) and one free magnon is smaller than the binding energy of a single polaron. Hence the peak corresponding to formation of the 2M1H state will be at a smaller frequency. Secondly, the amplitude of this peak is expected to be small due to the reduced probability of photoexciting an atom in the vicinity of a large composite object.

We find a general parity selection rule which makes the pair state dark or bright depending on the momentum given to the photoexcited atom. For fermions, this implies that we need a two-photon process in order to access finite momenta where the pair is a bright state. For bosons, the consequencues are more significant, as there seems to be no region where the anti-bound state exists and is bright simultaneously. To circumvent this problem, we propose an extended protocol which makes the anti-bound state bright under photoexcitation.

The spectroscopic approach to detect kinetically induced bound states can also be em-

ployed in other geometries which present kinetic frustration such as the square geometry with a perpendicular static magnetic flux [70]. In this situation, we expect the appearance of well resolved peaks below the free magnon dispersion relation in both fermionic and bosonic systems. Moreover, our results could be extended to study the Nagaoka's polaron from a spectroscopic approach. When the size of Nagaoka's polaron is large, we expect formation of collective modes associated with disturbances of the ferromagnetic background surrounding a doublon which could be associated with the formation of surface modes. We envision that the spectroscopic approached proposed in this work could shed some light on the collective excitations of the Nagaoka's polaron. Besides, in this work we have only focused on the single polaron spectroscopy. However, analysis of interactions between polarons is also an interesting research avenue [71, 88–92] and we plan to address this question in future publications.

# Acknowledgements

We acknowledge useful discussions with Waseem S. Bakr, Utso Bhattacharya, Annabelle Bohrdt, Anant Kale, Youqi Gang, Markus Greiner, Fabian Grusdt, Lev Haldar Kendrick, Wen-Wei Ho, Atac Imamoglu, Martin Lebrat, Max L. Prichard, Benjamin M. Spar, Muqing Xu, Zoe Z. Yan. Tensor Network computations have been performed using TeNPy [93].

**Funding information**  I.M. and E.D. acknowledge support from the the SNSF Project No. 200021_212899, the Swiss State Secretariat for Education, Research and Innovation (contract number UeM019-1) and the ARO Grant No. W911NF-20-1-0163. The work of C. W. and K.S is funded by the Cluster of Excellence "CUI: Advanced Imaging of Matter" of the Deutsche Forschungsgemeinschaft (DFG)—EXC 2056 —Project ID No. 390715994 and by the European Research Council (ERC) under the European Union's Horizon 2020 research and innovation program under Grant Agreement No. 802701.

# A   Scaling analysis

As explained in the main text, spectroscopy measurements can reveal the appearance of hole-magnon bound states. These states are revealed by the presence of peaks at lower frequencies than the characteristic frequency of the free-magnon propagation. Another interesting difference between bound states and free particles is the scaling of the weight of the peak with the system size and particle number. In these sections we present a detailed finite size and finite number scaling.

## A.1   Finite size scaling

To see hole-magnon bound states we need to photoexcite a magnon close to a hole. In this way we have a finite overlap with the hole-magnon bound state wavefunction. Thus the weight of the peak is directly proportional to the density of holes in the system. Therefore for a fixed number of holes we expect a weight proportional to the inverse of the system size. However the free-magnon wavefunction is independent of the number of holes and we expect the weight of the corresponding peak to be constant with system size, see Fig. 8. In our simulations we observe that the weight of the peak associated with the hole-magnon bound state decreases by increasing system size. We also observe that the weight of the free-magnon peak increases by increasing system size, denoting finite size effects in the free-magnon propagation.

For multi-particle bound states the weight of the respective peaks should decrease even faster than the inverse of the system size. In particular, for observing hole-hole-magnon bound states we need to photoexcite a magnon close to two holes. This probability scales with the density of holes squared. Thus for fixed number of holes the weight of this peak decays faster than the one associated with the hole-magnon bound state, see Fig. 9.

## A.2   Finite density scaling

The finite size scaling can be used to verify the nature of the different peaks observed in photoexcitation measurements. Larger multi-body states correspond to peaks with weights decaying faster with the sytem size. However, the weights also growth faster with the number of holes in the system. This points out that for fixed system sizes and increasing hole density, multi-body bound states dominate the photoexcitation spectrum over smaller composite

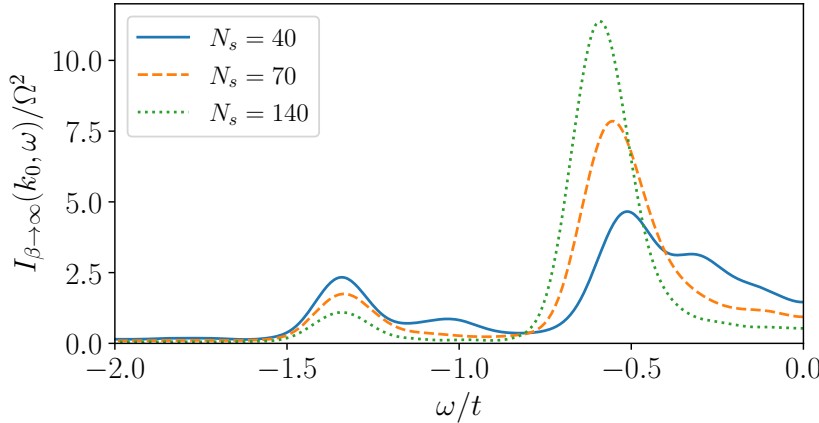

Figure 8: Photoexcitation spectrum at momentum $k_m = k_0$ as a function of the probed frequency $\omega$ for a fermionic system in a triangular ladder for different system sizes with $N_h = 4$ and interaction strength $U_{\uparrow\downarrow}/t = 20$.

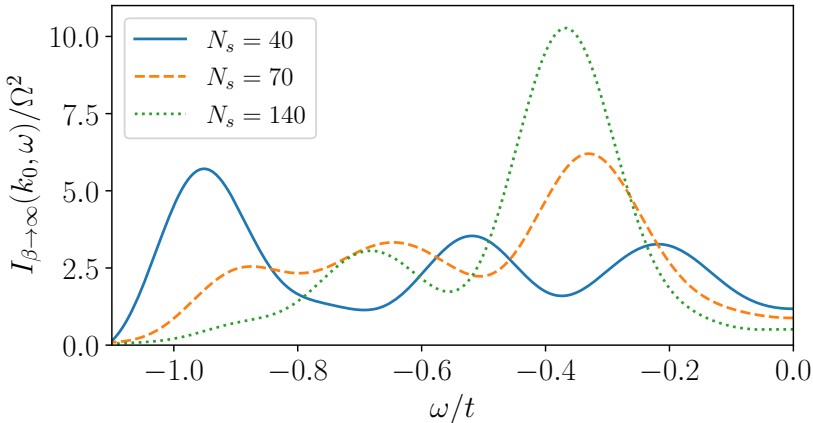

Figure 9: Photoexcitation spectrum at momentum $k_m a = \pi$ as a function of the probed frequency $\omega$ for a fermionic system in a triangular ladder for different system sizes with $N_h = 4$ holes and interaction strength $U_{\uparrow\downarrow}/t = 20$. At small system sizes we observe three main peaks corresponding to free-magnon, hole-magnon bound state and hole-hole-magnon bound state, from large to small frequencies. As the system size is increased the weight of the hole-hole-magnon peak decays faster than the other two.

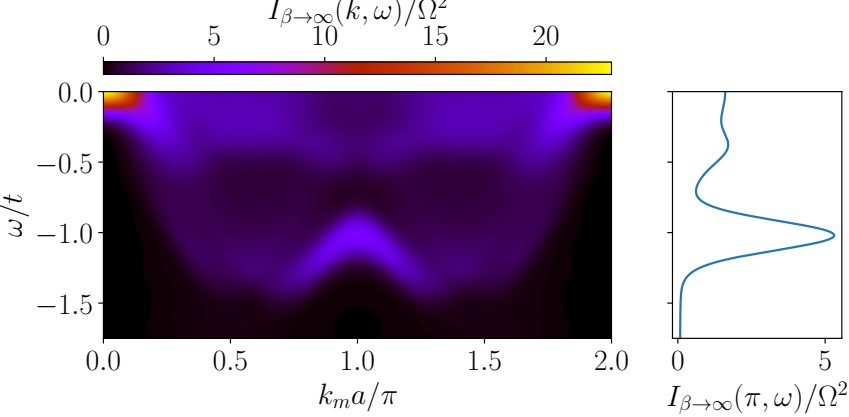

Figure 10: Photoexcitation spectrum as a function of the probed frequency $\omega$ and momentum $k$ for a fermionic system in a triangular ladder of $N_s = 70$ sites with $N_h = 10$ holes and interaction strength $U_{\uparrow\downarrow}/t = 20$. On the right we show a cut of the photoexcitation spectrum at the edge of the Brilluoin zone $ka = \pi$. Most of the weight is concenctrated at lower frequencies where trimers are formed.

objects. In Fig. 10 we show the photoexcitation spectrum for a system with a hole density $n_h = 1/7$. In this regime of large number of holes the trimer state formed by two holes and one magnon dominates the spectrum at the edge of the Brilluoin zone $ka = \pi$. Hole-magnon bound states have a smaller contribution due to their linear density scaling with hole density.

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
