# Peer review of "Exploring kinetically induced bound states in triangular lattices with ultracold atoms: spectroscopic approach"

_SciPost Physics_

## Round 1 · Referee Report · Anonymous (Referee 1) · 2024-1-24

Strengths

  1. The contribution is original and creative
  2. The main ideas are presented in a clear way
  3. The numerical results are robust and they are supported by convincing arguments.

Weaknesses

  1. The references cited by the authors are not always the most adequate.
  2. There are some minor issues of presentation that are listed in my report.

Report

The primary aim of this study is to investigate the application of spectroscopic techniques for ultracold atoms in detecting kinetically induced magnetic polarons within weakly doped Mott insulators characterized by frustrated triangular geometries.

The authors examine fermionic and bosonic Hubbard systems of spin-½ particles moving in a triangular lattice. They employ the Density Matrix Renormalization Group (DMRG) method to compute the dynamical spin structure factor, accessible through a two-photon Raman excitation.

The manuscript is well-crafted. The proposed strategy for detecting various many-body bound states of holes and magnons is original and supported by a combination of controlled approaches, including exact solutions of three and two-body bound states, along with DMRG calculations. The authors meticulously address details of their numerical results, providing a clear justification for understanding them in simple terms.

Additionally, the manuscript delves into experimental considerations, where the authors assert the realism of their proposal.

Given these considerations, I recommend the publication of this manuscript in Sci Post, contingent upon the authors considering the optional suggestions outlined below.

Abstract:

"In triangular-type lattices for large U/t, magnetism gets enhanced by doping away from n = 1 because kinetic energy of dopants can be lowered through developing magnetic correlations."

Note that this is true only for one type of doping (electrons or holes) depending on the sign of the hopping integral. Since this statement appears in the abstract, the authors may consider clarifying this point for general readers.

The prediction of magnetic polarons in the triangular lattice is rooted in the work referenced as [69] in the manuscript. This reference dates back to 2018, predating the three works cited in the abstract, which were published within the last two years. While I recognize that the cited works provide experimental confirmation of the prediction, it appears fitting to include a citation to the pioneering work that initially foresaw the phenomenon in the abstract. Incidentally, the formation of a bound state between two holes and a magnon (bipolaron) was also predicted in Ref. [69].

Section 5.1.2

The method described in section 5.1.2 (applying the Lanczos method to a target state to compute dynamical correlation functions) was originally introduced in 1988 (Phys. Rev. B 38, 11766 – Published 1 December 1988). The reference [86] cited by the authors is from 2012.

Caption of Figure 1

"Dashed line corresponds to the free ↑-fermion dispersion relation." It is more appropriate to say that the dashed line is the free magnon dispersion (magnons are bosonic modes). The "free ↑-fermion dispersion relation" is the one obtained by diagonalizing the kinetic energy term of Eq. (1).

Right above section 6.1

"However, the total quasi-momentum Q = km − kh of the hole-magnon bound state is not a good quantum number for a system with more than a single hole." This sentence is a bit confusing because the total momentum of the state is still a good quantum number. The authors indicate that the two states are mixed because of the presence of a second hole. What is the symmetry that protects the crossing of the two bands at Q=pi,0 in the exact solution of the two-body problem?

Eq. (15)

J_{\perp} should be equal to – 4t^2/U_{\up \down} because of the bosonic character of the spin 1/2 particles.

Eq. (15)

The last term of the Hamiltonian violates the conservation of N_{up}-N_{dw} (I guess that the spin of the annihilation operator should be \bar{\sigma} instead of \sigma).

5th line below Eq. (15)

There is a typo in the sentence: "The first one allows to holes to hop…"

Conclusions and Outlook

There is a typo near the end:

"We envision that the spectroscopic approached proposed…"

Requested changes

The proposed modifications outlined in my report are merely suggestions. Authors have the discretion to adopt these recommendations if they deem them suitable.

Attachment

  • validity: top
  • significance: high
  • originality: top
  • clarity: high
  • formatting: good
  • grammar: good

Author:  Ivan Morera Navarro  on 2024-02-16  [id 4313]

(in reply to Report 2 on 2024-01-24)

We thank there Referee for their careful and thoughtful assessment of our manuscript. Their
suggestions has helped us to improve the presentation and content of our work. Moreover, we are pleased to see that the Referee recommends publication in Sci Post. In this response, we address the comments by the Referee.

Abstract:

We agree with the Referee and we now explicitly state the types of magnetism obtained depending on doping: “In triangular-type lattices for large $U/t$ and $t>0$, antiferromagnetism (ferromagnetism) gets enhanced by doping away from $n=1$ with holes (doublons) because kinetic energy of dopants can be lowered through developing magnetic correlations…”

Moreover, we have included Reference [69] in the abstract.

Section 5.1.2:

We thank the Referee for providing these valuable references and we have included now in our manuscript.

Caption of Figure 1:

We agree with the Referee and we have changed the text in the new version of the manuscript.

Right above section 6.1:

The symmetry that protects the crossing in the exact solution of the two-body (hole-magnon) problem is translational invariance. These two bands have different total quasi-momentum of the hole-magnon bound state and therefore, they cannot be connected by the Hamiltonian. One solution has Q=km-kh and the other one has Q=km+kh. However, if a second hole is added, then, the hole-magnon quasi-momentum is no longer a conserved quantity. Therefore, there can be a matrix element between the two branches provided an extra hole to exchange momenta, such that the total quasi-momentum (hole+hole+magnon) is conserved.

To avoid confusion we now state: “However, the quasi-momentum Q = km − kh of the hole-magnon bound state…”. Moreover, we have changed the discussion in Section 6.1 clarifying these concepts.

Eq. (15):

We thank the Referee for spotting these typos and we have changed the Equation accordingly.

Typos:

We have corrected the typos in the text.

---

## Round 1 · Referee Report · Anonymous (Referee 1) · 2024-1-24

Strengths

  1. The contribution is original and creative
  2. The main ideas are presented in a clear way
  3. The numerical results are robust and they are supported by convincing arguments.

Weaknesses

  1. The references cited by the authors are not always the most adequate.
  2. There are some minor issues of presentation that are listed in my report.

Report

The primary aim of this study is to investigate the application of spectroscopic techniques for ultracold atoms in detecting kinetically induced magnetic polarons within weakly doped Mott insulators characterized by frustrated triangular geometries.

The authors examine fermionic and bosonic Hubbard systems of spin-½ particles moving in a triangular lattice. They employ the Density Matrix Renormalization Group (DMRG) method to compute the dynamical spin structure factor, accessible through a two-photon Raman excitation.

The manuscript is well-crafted. The proposed strategy for detecting various many-body bound states of holes and magnons is original and supported by a combination of controlled approaches, including exact solutions of three and two-body bound states, along with DMRG calculations. The authors meticulously address details of their numerical results, providing a clear justification for understanding them in simple terms.

Additionally, the manuscript delves into experimental considerations, where the authors assert the realism of their proposal.

Given these considerations, I recommend the publication of this manuscript in Sci Post, contingent upon the authors considering the optional suggestions outlined below.

Abstract:

"In triangular-type lattices for large U/t, magnetism gets enhanced by doping away from n = 1 because kinetic energy of dopants can be lowered through developing magnetic correlations."

Note that this is true only for one type of doping (electrons or holes) depending on the sign of the hopping integral. Since this statement appears in the abstract, the authors may consider clarifying this point for general readers.

The prediction of magnetic polarons in the triangular lattice is rooted in the work referenced as [69] in the manuscript. This reference dates back to 2018, predating the three works cited in the abstract, which were published within the last two years. While I recognize that the cited works provide experimental confirmation of the prediction, it appears fitting to include a citation to the pioneering work that initially foresaw the phenomenon in the abstract. Incidentally, the formation of a bound state between two holes and a magnon (bipolaron) was also predicted in Ref. [69].

Section 5.1.2

The method described in section 5.1.2 (applying the Lanczos method to a target state to compute dynamical correlation functions) was originally introduced in 1988 (Phys. Rev. B 38, 11766 – Published 1 December 1988). The reference [86] cited by the authors is from 2012.

Caption of Figure 1

"Dashed line corresponds to the free ↑-fermion dispersion relation." It is more appropriate to say that the dashed line is the free magnon dispersion (magnons are bosonic modes). The "free ↑-fermion dispersion relation" is the one obtained by diagonalizing the kinetic energy term of Eq. (1).

Right above section 6.1

"However, the total quasi-momentum Q = km − kh of the hole-magnon bound state is not a good quantum number for a system with more than a single hole." This sentence is a bit confusing because the total momentum of the state is still a good quantum number. The authors indicate that the two states are mixed because of the presence of a second hole. What is the symmetry that protects the crossing of the two bands at Q=pi,0 in the exact solution of the two-body problem?

Eq. (15)

J_{\perp} should be equal to – 4t^2/U_{\up \down} because of the bosonic character of the spin 1/2 particles.

Eq. (15)

The last term of the Hamiltonian violates the conservation of N_{up}-N_{dw} (I guess that the spin of the annihilation operator should be \bar{\sigma} instead of \sigma).

5th line below Eq. (15)

There is a typo in the sentence: "The first one allows to holes to hop…"

Conclusions and Outlook

There is a typo near the end:

"We envision that the spectroscopic approached proposed…"

Requested changes

The proposed modifications outlined in my report are merely suggestions. Authors have the discretion to adopt these recommendations if they deem them suitable.

Attachment

---

## Round 1 · Referee Report · Anonymous (Referee 2) · 2024-2-8

Strengths

  1. The main ideas are presented in an intuitive way without unnecessary jargon and only using general physical arguments.
  2. The proposed experimental protocols are conveyed in simplistic pictorial ways that are (mostly) easy to understand.
  3. The authors assess the feasibility of the proposed approaches in actual experiments taking into account that experiments work at nonzero temperatures.

Weaknesses

  1. In several instances, it is not clear how the authors arrive at a specific result given in an equation.
  2. Although the authors have a section on their numerical methods, they only specify in that section what they are used for. It would be helpful if the authors at the presentation of their results refer back to the numerical methods section and state clearly what method has been used for what.

Report

The aim of the paper is to define spectroscopic protocols that access bound hole-magnon states in frustrated magnetic lattices.

The paper is well-written, has novel and interesting results, and mostly present them without unnecessary jargon. They also provide evidence that the proposed approaches can be realistically implemented experimentally by examining the robustness to nonzero temperatures.

In total, the manuscript definitely merits publication in SciPost Physics. There are, however, are few issues in the presentation and important details that remain unclear throughout the manuscript. I, therefore, strongly encourage the authors to address the requested changes below, after which I can whole-heartedly support its publication in SciPost Physics.

Requested changes

Major comments: 1. In the beginning of Sec. 3, the authors very briefly describe the kinetic frustration effect that is highly important for the system in consideration. However, this is not described in much detail, which is quite unfortunate. A cartoon of the lack of interference would be appropriate, even though this has appeared in some of the given references.

  1. Also in Sec. 3, the authors describe the process of exciting magnons in the vicinity of holes, and state that this happens with "finite probability". I think it would be worth pointing out, what it scales like. This would presumably scale with the hole density, $n_h$.

  2. On page 5, the authors state: "The probability of this event is proportional to the density of hole-magnon bound states, given by min($N_h,N_\uparrow$)". These should, however, be densities, right? Else the notion of probability is out of the window. So min($N_h,N_\uparrow$) $\to$ min($n_h,n_\uparrow$) seems appropriate.

  3. Please move the figures such that they appear more or less with the text that refers to them. This is particularly extreme with Fig. 1, where there are very detailed references to the figure on page 5, while the figure only appears on page 9.

  4. The section "Numerical Methods" describes how the dynamical spin structure factor can be computed at ${\bf zero}$ temperature. However, the authors also present results at nonzero temperatures, Fig. 4, and it is completely unclear to me how they get those results.

  5. It is completely unclear to me, how the authors arrive at the approximate bound state dispersion relations in Eq. (13). This should be explained a lot better.

  6. In Sec. 6.1, the authors explain an avoided crossing happening around $k_m = \pi$. Firstly, however, this avoided crossing is basically completely hidden by the dotted orange line in Fig. 1(d). One basically has to know that it is there to see it. Secondly, the notation for the momentum states is quite confusing. A hole state at momentum $k_h$ is actually a state with momentum $-k_h$. Therefore, the states $|\Phi^{+}_{k_m - k_h}\rangle |-k_h\rangle$ actually has total momentum $k_m$. That is pretty unclear from the presentation, and more care should be taken here, I think.

  7. The authors explain that experiments can get a resolution of order $t$. However, the results in Fig. 3 to distinguish the hole-2 magnon state from the hole- 1 magnon state requires a resolution one order of magnitude better ($0.1t$). I think the authors should address the feasibility to see this splitting in actual experiments.

  8. Also related to Fig. 4. This has a lot less resolution than the previous spectra. This deserves a comment and an explanation.

  9. Equation (16) is hardly explained. Please spend some time explaining the different factors in the expression. Here, I think it is pretty unclear what "n" refers to e.g. Also, below this equation the hole dispersion is denoted $\epsilon$ instead of $w_h$.

  10. Sec. 7.2: Pump and probe protocol. This idea is a bit complex, but very nice. It is not clear, however, how the process goes in more detail. The authors e.g. state that "...Once the holes are accelerated we photoexcite". What do you mean precisely? How long do you wait, how long should one wait, and how does this influence the signal.

  11. The conclusion section is to a large extend a direct copy of parts of the introduction. I think this should be avoided. The expression min($N_h, N_\uparrow$) instead of the more appropriate min($n_h, n_\uparrow$) again appears.

  12. In the abstract, the authors write: "Snapshots of many-body states obtained with quantum gas microscopes [1–3] demonstrated existence of magnetic polarons around dopants at temperatures that..." I think this is a confusing phrasing. The magnetic polarons include the dopant itself, so it is not forming "around dopants".

  13. In the list of references on magnetic polarons in square lattices in the introduction, the authors mainly refer to historical articles dating back at least to 2001. They actually only cite a single more recent paper, which is coauthored by one of the authors of the present manuscript. This is unfortunate. Either they should remove that reference, or they should include more of the many recent papers on this subject, trying to cover different techniques and results.

  14. The appearance of magnetic polarons in square lattices has recently been put into question via the so-called phase string effect. Therefore, strictly speaking I do not believe that there is any direct proof that the experimental findings in these systems show that magnetic polarons emerge. I believe this is relevant to comment on for two reasons: (1) They refer to magnetic polarons in square lattices, which may or may not exist. And (2) Is this relevant or not for the triangular lattice case? Here are some recent (and one more historical) reference on the subject: PRB 103, 035141 (2021); PRL 77, 25 (1996); PRX 12, 011062 (2022); PRB 92, 235156 (2015); PRB 107, 085112 (2023).

Minor comments

  1. In the introduction, the authors repeatedly misspell magnet as "magent".

  2. At the top of page 6, the authors use a notation diag(a). I presume this means a times the 2 by 2 identity matrix, but this is pretty unclear from the presentation.

  3. On the middle of page 7, the authors write: "By choosing an initial value of lambda $λ_0 = 0$..." Here they should delete "lambda".

  4. In the caption of Fig. 3, it says that $k_m = 29\pi a / N_s$, but $a$ is the lattice constant, so it does not have the right unit, as far as I can tell.

  5. There is a missing index after Eq. (15) on $S^{\bar{\uparrow}}$.

  • validity: high
  • significance: good
  • originality: good
  • clarity: good
  • formatting: perfect
  • grammar: excellent

Author:  Ivan Morera Navarro  on 2024-02-16  [id 4312]

(in reply to Report 3 on 2024-02-08)
Category:
remark
answer to question
correction

We thank the Referee for their detailed and insightful assessment of our manuscript, which has significantly contributed to improve the quality of our work. Moreover, we are pleased to see that the Referee recommends publication in Sci Post. Here we provide a detailed response to the Referee’s comments.

Major comments:

  1. We provide a description of the interference pattern in the triangular lattice: “When a hole propagates in a kinetically frustrated lattice, including a triangular lattice, it experiences kinetic frustration. This phenomenon can arise when a lattice exhibits closed loops composed of either an odd or even number of links. In this scenario, different quantum propagation paths of a single hole destructively interfere due to their opposite contribution to the energy when the hopping integral is positive (t > 0) [1, 76, 88–90].”

  2. We clarify that this probability scales with the hole density.

  3. We agree with the Referee and we now change it to $\min(N_h,N_{\uparrow})/N_s$.

  4. We have moved Figures such that they appear close to the text that refers them.

  5. We thank the Referee for pointing this out and we have now included an extra subsection “Two-body solution and temperature effects” where we explain how the results of Fig. 4 are obtained.

  6. We now explain how to get the dispersion relation and that Eq. 13 is a good approximation to our numerical results.

  7. We now explain that dotted orange line can only capture a direct crossing since it is obtained for a single hole and a single magnon. We explicitly specify the total momentum of the state in the new version of the manuscript: “Note that these states have a total quasi-momentum equal to $k_m$ since the state $|- k_h\rangle$ has a quasi-momentum $k_h$.”. Moreover, we emphasize our convention for the momenta of the holes in Sec. 5.3: ”Note that we define the hole state $|\vec{k}_h\rangle$ such that it exhibits a physical quasi-momentum $-\vec{k}_h$.”

  8. We now address the feasibility of observing the peak splitting in current experiments at the end of Sec. 6.2.

  9. We now explain that this Figure is obtained with a different technique and we provide details in Sec. 5.3.

  10. We have moved this equation to the new Section 5.3 where we explain in detail how to derive it.

  11. We now propose that photoexcitation should be performed immediately after acceleration to avoid effects from the external trap. “The holes should stay at finite quasi-momentum after the acceleration in a translational invariant system, but we propose to apply a two-photon Raman process immediately to avoid decoherence from an external trap.”

  12. We have eliminated the parts that are repeated from previous sections.

  13. We agree with the Referee and we have now changed the text: “Snapshots of many-body states obtained with quantum gas microscopes [2–4] demonstrated existence of magnetic polarons by revealing the magnetic correlations around dopants at temperatures that considerably exceed superexchange energy scale”.

  14. We now include more recent references on the subject of magnetic polarons in square lattices.

  15. We are grateful to the Referee for providing us with these valuable references, which we now include in the new version of our manuscript. We agree with the Referee that magnetic polaron formation in square lattice is a complicated topic. However, let us remark that by magnetic polaron formation we just want to emphasize the distortion of magnetic correlations upon carrier injection without going into details if the quasi-particle picture is broken. Early numerical studies (Phys. Rev. B 64, 024411 (2001)) showed that Nagaoka’s polaron appears for the square lattice in the regime of very strong interactions U/t>200. This underlines the difficulty of observing such an object in cold atom experiments, where reaching such extreme large values of interaction without penalizing the hopping is extremely difficult. However, let us remark, that Nagaoka’s polaron appears for a much moderate interaction regime in the triangular lattice due to the magnetic frustration of the antiferromagnetic order. This explains why the recent cold atom experiments [2-4] have been successful in identifying magnetic polarons in the triangular lattice.

Minor comments:

We thank the Referee for pointing out these typos and we have corrected them. Moreover, we now specify the notation diag(a): “with $\mathrm{diag}(\vec{k} \cdot \vec{\delta})$ a $N_s\times N_s$ diagonal matrix with entries $\vec{k}1 \cdot \vec{\delta},\dots,\vec{k}$} \cdot \vec{\delta and similarly for $\epsilon_{\vec{k}}$.”

---

## Editorial Decision

resubmitted